# CD105 Is Expressed in Ovarian Cancer Precursor Lesions and Is Required for Metastasis to the Ovary

**DOI:** 10.3390/cancers11111710

**Published:** 2019-11-02

**Authors:** Shoumei Bai, Wanhong Zhu, Lan Coffman, Anda Vlad, Lauren E. Schwartz, Esther Elishaev, Ronny Drapkin, Ronald J. Buckanovich

**Affiliations:** 1Magee-Womens Research Institute, Department of Obstetrics, Gynecology and Reproductive Sciences, University of Pittsburgh, Pittsburgh, PA 15213, USA; bais@mwri.magee.edu (S.B.); coffmanl@mwri.magee.edu (L.C.); avlad@mwri.magee.edu (A.V.); elisex@upmc.edu (E.E.); 2Division of Hematology-Oncology, Department of Internal Medicine, University of Michigan, Ann Arbor, MI 48109, USA; wanhongz@umich.edu; 3Department of Pathology and Laboratory Medicine, Perelman School of Medicine, University of Pennsylvania, Philadelphia, PA 19104, USA; 4Penn Ovarian Cancer Research Center, Department of Obstetrics and Gynecology, Perelman School of Medicine, University of Pennsylvania, Philadelphia, PA 19104, USA; rdrapkin@pennmedicine.upenn.edu; 5Hillman Cancer Center, Division of Hematology Oncology, University of Pittsburgh Medical Center, Pittsburgh, PA 15232, USA

**Keywords:** High-grade serous ovarian cancers, hematogenous spread, endoglin (CD105), TGF-β, TRC105, serous tubal intraepithelial carcinoma (STIC)

## Abstract

Most high-grade serous ovarian cancers (HGSCs) initiate from the fallopian tube epithelium and then metastasize to the ovary and throughout the abdomen. Genomic analyses suggest that most HGSCs seed the ovary prior to abdominal dissemination. Similarly, animal models support a critical role for the ovary in driving abdominal dissemination. Thus, HGSC cell recruitment to the ovary appears to be a critical component of HGSC cell metastasis. We sought to identify factors driving HGSC recruitment to the ovary. We identified CD105 (endoglin, or ENG, a TGF-β receptor family member) as a mediator of HGSC cell ovarian recruitment. We found that CD105 was expressed on both serous tubal intraepithelial carcinoma (STIC) cells (STICs-HGSC precursors in the fallopian tube epithelium) and HGSC cells. Using data from The Cancer Genome Atlas (TCGA) and the Cancer Cell Line Encyclopedia (CCLE), we showed that high CD105 expression by HGSC cells correlated with a metastatic signature. Furthermore, intravenous injection of CD105^(+)^ HGSC tumor cells, but not CD105^(−)^, resulted in ovarian-specific metastasis and abdominal dissemination of disease. CD105 knockdown or blockade with a clinically relevant CD105-neutralizing mAb (TRC105), inhibited HGSC metastasis, reduced ascites, and impeded growth of abdominal tumor nodules, thereby improving overall survival in animal models of ovarian cancer. CD105 knockdown was associated with a reduction in TGF-β signaling. Together, our data support CD105 as a critical mediator of ovarian cancer spread to the ovary and implicate it as a potential therapeutic target.

## 1. Introduction

Approximately 80% of ovarian cancer patients present with late-stage disease, metastatic beyond the ovary. The majority of deaths caused by ovarian cancer relate to metastatic disease, particularly to diffuse abdominal spread resulting in tumor ileus and bowel obstruction. Thus, metastasis in ovarian cancer is a major clinical problem, and inhibiting metastasis is a major unmet clinical need.

While it was initially presumed that ovarian cancer metastasis was due to direct shedding from the ovary into the abdominal cavity, increasing evidence suggests that ovarian cancer can metastasize via the vasculature. Ovarian tumor cells can survive and develop secondary metastases when introduced into the blood through peritoneovenous shunting [1]. Circulating tumor cells can be identified in the bloodstream of many patients with ovarian cancer [2,3,4,5], and lymphovascular spread and metastasis is well established [6,7,8,9]. Clinical studies indicate hematogenous metastasis can occur independently of peritoneal spread and is associated with poorer survival outcome [10].

Recent laboratory studies support a more prominent role for hematogenous spread of ovarian cancer. Perhaps the best illustration of hematogenous spread of ovarian cancer is a parabiosis model in which two mice share a blood stream. Using this model, Sood and colleagues demonstrated that ovarian tumor cells from the ovary of one mouse could spread via the vasculature to the parabiosed partner [11].

To better study the hematogenous spread of ovarian cancer, we recently developed several ovarian cancer metastasis models [12,13]. Ovarian cancer cells, whether injected intravenously or subcutaneously in the axilla, demonstrate their ability to metastasize hematogenously to the ovary. Suggesting a critical role for the ovary as a driver of intraperitoneal spread, we and others have shown that removal of the ovary in mouse models of ovarian cancer significantly reduces intra-abdominal tumor burden [13,14,15]. These data, combined with the observation that most high-grade serous ovarian cancers (HGSCs) likely initiate as serous tubal intraepithelial carcinomas (STICs) in the fallopian tube epithelium [15,16,17], suggest that the ovary may be a primary metastatic site. Once cancer cells are in the ovary, their ability to spread throughout the abdomen is enhanced [12,13].

Understanding the factors that recruit HGSC cells to the ovary could be critical to blocking metastasis. While the ovary is thought to attract malignant cells through production of inflammatory cytokines and hormonal signaling, little is known about intrinsic factors on the tumor cells that are involved in spread to the ovary [18]. A better understanding of the factors that regulate ovarian homing of HGSC cells could lead to the identification of therapeutic targets for preventing ovarian metastasis.

A candidate regulator of ovarian homing is CD105 (endoglin, ENG). CD105 is a type I glycoprotein located on the surface of cells that serves as a TGF-β co-receptor [19,20]. TGF-β receptors activate downstream factors SMAD2/3 to translocate to the nucleus and control gene expression. In cancer cells, this pathway can promote uncontrolled proliferation, invasion, and metastasis [7,21,22]. CD105 expression has been associated with poor prognosis in several cancers, including ovarian cancer [23,24,25], and high CD105 expression is associated with increased levels of metastasis in oral, lung, liver, and pancreatic cancer [26,27,28,29].

Much of the study of CD105 in cancer has related to its role in tumor vasculature and neo-angiogenesis [28,30,31,32]. Less is known about its function on cancer cells. CD105 expression in ovarian cancer cells has been associated with chemotherapy resistance [24,25], but its function is otherwise unclear.

Here, we report that CD105 was expressed in fallopian tube STIC lesions. Using an intravenous injection model of ovarian cancer hematogenous spread, we found that CD105^(+)^ ovarian cancer cells, versus CD105^(−)^ cells, preferentially colonized the ovary and induced abdominal metastasis. CD105 knockdown in tumor cells resulted in reduced ovary colonization and curtailed abdominal spread. Importantly, we found the anti-CD105 antibody TRC105 effectively reduced hematogenous ovarian metastasis in mice and improved animal survival. This study supports CD105 as a therapeutic target in ovarian cancer therapy and, potentially, in its prevention.

## 2. Results

### 2.1. CD105 Is Expressed in Both STICs and Ovarian Cancer Cells That Metastasize to the Ovary

The TGF-β pathway is strongly linked with metastasis [33,34,35]. CD105 is a TGF-β co-receptor that has been linked with metastasis and cancer stem cells [23,24,36,37]. We therefore hypothesized that CD105 could play a role in the metastasis of HGSC cells from the fallopian tube epithelium to the ovary. We evaluated CD105 expression in fallopian tube STICs. Fallopian STIC lesions were identified histologically and confirmed by p53 IHC. IHC analysis of STIC lesions confirmed moderate to strong CD105 expression in eight of eight lesions evaluated (Figure 1, Appendix A). However, CD105 was not expressed on all cells in the STIC. Strong staining of CD105 was also observed in stroma surrounding some STICs (Figure 1A and Appendix A).

We previously identified ovarian cancer cell lines and primary ovarian cancer cells that can hematogenously metastasize to the mouse ovary and then disseminate throughout the abdomen [12]. We expanded these studies and evaluated CD105 expression in six ovarian cancer cell lines (OVCAR3, Kuramochi, CAOV3, SKOV3, Hey1, and A2780) and three primary ovarian cancer patient samples. Four of six cell lines and three of three patient samples demonstrated significant CD105 expression. All seven samples with CD105 expression demonstrated the ability to metastasize to the ovary following intravenous (IV) or subcutaneous (SQ) injection in mice (Figure 1B). Two cells lines (CAOV3 and A2780) did not express CD105, and neither of these cell lines metastasized to the ovary in either the SQ or the IV model. 

### 2.2. Elevated CD105 Expression on Ovarian Cancer Cells Enriches Functional Networks Associated with Vascular Invasion and Metastasis

To gain an understanding of the significance of CD105 expression on ovarian cancer cells, we first analyzed CD105 gene expression in RNA-Seq data obtained from 307 ovarian cancer samples recorded in the TCGA dataset (OVCA-TCGA). Seventy-six of these tumors were defied as CD105-high based on quintile distribution (Z > Q3 Figure 2A). Gene expression profiles were compared between CD105-high and the remaining tumor samples Z ≤ Q3, Figure 2A). Genes that were positively or negatively correlated significantly with CD105-high expression (*p* < 0.05, Pearson score *r* > 0.3 or *r* < −0.2) were used for gene enrichment analysis using Crosstalker. Interestingly, many of the enriched networks, such as ‘cell-surface interaction at the vascular wall’, ‘Extracellular Matric (ECM) organization’, and ‘integrin cell surface interactions’, are linked to critical steps of hematogenous metastasis (Figure 2B–D) [38,39,40,41].

While TCGA samples are enriched for cancer cells, they still represent whole tumor expression and cannot directly be linked to CD105 function on tumor cells as CD105 is also expressed on tumor endothelial cells and cancer-associated mesenchymal stem cells [30,42]. To determine whether these results relate to cancer cell CD105 expression, we repeated this analysis using ovarian cancer cell line RNA-Seq data from CCLE (OVCA-CCLE, https://portals.broadinstitute.org/ccle/). Thirteen of 52 ovarian cancer cell lines were CD105-high (Z > Q3). Gene expression profiles were compared between CD105-high and CD105-low (Z < Q1, Figure 2E) and analyzed using Crosstalker as above. Strongly suggesting the TCGA results were related to cancer cell CD105 expression, we observed a strikingly similar result, with ECM organization, cell surface interaction at the vascular wall, and integrin signaling all being upregulated in the CD105-high group. Consistent with CD105 being a TGF-β co-receptor, and supporting the fidelity of the analysis, SMAD2/3 signaling was also enriched (Figure 2F,G).

### 2.3. CD105^(+)^ but Not CD105^(−)^ Cells Hematogenously Metastasize to the Ovary 

We next tested the role of CD105 in the hematogenous spread of ovarian cancer using the SKOV3 ovarian cancer cell line. We chose it as it has very distinct CD105^(+)^ and CD105^(−)^ cell populations. SKOV3 cells were grown subcutaneously (SQ) in the flank of a mouse or injected intravenously to generate ovarian cancer metastases and ascites. We then evaluated CD105 expression in (i) SKOV3 cells in culture, (ii) SKOV3 SQ tumor, (iii) IV injection-related SKOV3 ovarian metastases, and (iv) IV injection-associated SKOV3 ascites. Interestingly, while SKOV3 cells in culture and in the SQ tumor remained 30–40% CD105^(+)^, ovarian metastases and metastatic ascites cancer cells were ≥95% CD105^(+)^ (Figure 3A).

As cancer stem cells (CSCs) have been linked with metastasis, we evaluated expression of several reported CSC markers. We first evaluated the expression of the CSC marker ALDH [12,43]. CD105^(+)^ cells were also ALDH^(+)^ cells in both bulk SKOV3 and SQ tumors. Approximately 62% of CD105^(+)^ cells were ALDH^(+)^ cells in ovarian metastasis, and 34% in abdominal metastasis (Figure 3B). We evaluated five additional cell surface markers that have been linked with “stemness,” including CD44, CD73, CD90, CD133, and SSEA. CD45 was used as a negative control. While four of these markers were detected in SKOV3 cells in culture (CD44, CD73, CD90, and SSEA4), none were detected in the metastatic cells (Figure 3C,D). CD133 and CD45 (negative control) were not detected in either SKOV3 cells in culture or ovarian metastasis (Appendix A).

To directly evaluate the role of CD105 ovarian cancer cell metastasis to the ovary, we double-purified CD105^(+)^ and CD105^(−)^ cells using magnetic bead separation followed by FACS sorting. The growth rate of sorted CD105^(+)^ and CD105^(−)^ cells were comparable as assessed during a 9 day cell culture (Appendix A). Purified CD105^(+)^ and CD105^(−)^ cells were injected by IV into mice. Lung metastases were observed from both CD105^(+)^ (62%; 5/8) and CD105^(−)^ cells (80%; 4/5). However, while CD105^(−)^ cells failed to colonize the ovary or develop intra-abdominal metastases (Figure 4A), CD105^(+)^ cells generated ovarian metastasis in 87.5% (7/8) of the mice within 2 months (Figure 4A). Four out of seven mice bearing ovarian tumors developed ascites, three of which also developed liver metastasis. Interestingly, mice injected with CD105^(−)^ tumor cells were grossly healthy at day 80, whereas all mice injected with CD105^(+)^ cells died of metastasis within 2 months (Figure 4B).

Evaluation of the tumor cell surface-marker expression in the ascites generated from CD105^(+)^ cell injections demonstrated expression in >98% CD105^(+)^/CD73^(−)^/CD44^(−)^ of cells (Figure 4C). Analysis of lung nodules derived from CD105^(−)^ cells demonstrated that 40% of the cells were CD105^(+)^; however, unlike those found in the metastatic sites, these were CD44^(+)^ CD73^(+)^ cells (Figure 4C). There was a small population of CD105^(+)^ CD44^(−)^ CD73^(−)^ cells. The origin of these cells, whether related to trace CD105^(+)^ contaminating cells, cellular differentiation/de-differentiation, or metastasis from the ovary back to the lung, remains to be determined.

### 2.4. CD105 as a Therapeutic Target for Ovarian Cancer Metastasis 

To confirm the role of CD105 in ovarian metastasis and evaluate it as a therapeutic target, we first tested the impact of CD105 knockdown. FACS-isolated CD105^(+)^ SKOV3 cells were transduced with CD105 shRNA or scrambled shRNA control. Significant reduction in the CD105 mRNA level was seen in two CD105 shRNAs (shCD105-1 or shCD105-2) by qRT-PCR (Figure 4D). Reduction of CD105 protein levels was confirmed by flow cytometry analysis (Figure 4E). In vitro analysis of CD105 knockdown cells suggests that CD105 knockdown does not significantly affect cell growth (Appendix A). The shCD105 cells and scrambled control cells were then injected by IV into mice. All mice were euthanized and evaluated for metastases when two mice that had been injected with the scrambled shRNA control cells died of disease 4 weeks after injection. The two mice that died were carrying a large ovarian tumor burden and ascites. Ovarian tumors were observed in all mice with scrambled shRNA control (5/5, 100%, Figure 4D). The shCD105 tumor cells failed to colonize the ovary in either shCD105-1 or shCD105-2 lines at the time of euthanasia. Interestingly, intraparenchymal metastatic tumors in the uterine horn were observed in two mice injected with shCD105-1, which retains more CD105 expression than that of the shCD105-2 knockdown (Figure 4C). 

As noted above, SKOV3, while commonly used as a representative of ovarian cancer, is not a high-grade serous cell line [43]. To extend the role of CD105 in ovarian cancer metastasis to high-grade serous ovarian cancer cells, we introduced shRNA (shCD105-1and-2) to OVCAR3, an HGSC cell line that is 100% CD105^(+)^ by FACS analysis. Reductions in both CD105 mRNA and protein levels were confirmed by qRT-PCR and FACS analysis, respectively (Figure 5Ai–ii). Once again, knockdown of CD105 in OVCAR3 did not significantly inhibit cell growth (Appendix A). 

We next tested the impact of CD105 inhibition on migration potential using the ECMatrix cell invasion assay. The shCD105 lines showed a 7–75% reduction of cell migration compared with scrambled control cells (Figure 5B). We then used these cells to repeat hematogenous metastasis studies. The shCD105 and scrambled control OVCAR3 cells were injected by IV into immune-deficient mice (*n* = 7 each of shCD105-1, shCD105-2, and scrambled control). Animals either died of disease or were euthanized when they lost >20% body weight. The shCD105 groups demonstrated a significant advantage in survival (*p* = 0.007). While 100% of control mice developed ovarian metastases and died of disease by day 64, two out of the seven mice in the shCD105-2 group (most effective CD105 knockdown) were free of ovarian cancer metastases 4 months after cell injection. The shCD105-1 group had an intermediate phenotype. All animals in this group eventually developed ovarian tumors and died by day 100 (Figure 5C).

We also tested the effect of CD105 knockdown on direct abdominal tumor cell spread of OVCAR3 cells. We injected shCD105-1 and -2 and scrambled control OVCAR3 cells intraperitoneally (IP) into SCID mice (*n* = 5 each). Mice injected with scrambled control shRNA developed extensive metastatic disease within 2 weeks, requiring euthanasia. At this time, the shCD105-1 and -2 groups appeared grossly healthy. Both shCD105-1 and -2 groups were euthanized 1 week later to allow a comparison of tumor nodules. Although tumors were allowed an additional week to implant, shCD105 cells developed significantly fewer nodules per mouse compared to controls (an average of 52 tumor nodules in both ShCD105 groups, as compared to an average of over 300 tumor nodules in the control group; Figure 5D and Appendix A).

### 2.5. CD105 Inhibition Downregulates TGF-β Signaling

CD105 serves as a TGF-β co-receptor. Activation of SMAD2/3 signaling is a known downstream effect of TGF-β signaling, and we observed enrichment of the SMAD2/3 signaling pathway in CD105-high ovarian cancer cell lines and tumors (Figure 2). We therefore assessed the impact of CD105 knockdown on the TGF-β/SMAD2/3 signaling pathway. CD105 shRNA knockdown in both SKOV3 and OVCAR3 cell lines resulted in downregulation of TGFBR2 and TGFBR3 in SKOV3 cells and downregulation of TGFBR1 and TGFBR2 in OVCAR3 cells (Figure 5E). Consistent with decreased TGFBR signaling, CD105 knockdown resulted in significant decreases in both total and phosphorylated SMAD2/3 (Figure 5F).

### 2.6. CD105 Neutralizing Antibody Therapy Inhibits Ovarian Metastasis

CD105 neutralizing antibody therapy (TRC105) is currently being tested in clinical trials (clinicaltrials.gov). To test the therapeutic potential of antibody-mediated targeting of CD105 in ovarian cancer, we first tested the impact of anti-CD105 antibody on cell migration in vitro. Treating OVCAR3 cells with CD105 neutralizing antibody (SN6) resulted in a 75% reduction of cell migration (Figure 6A). We next tested the impact of TRC105 on ovarian metastasis. Mice were treated with TRC105 IP (15 mg/kg) or PBS (control) for 3 days prior to IV injection of OVCAR3 cells. Subsequently, TRC105 or PBS was administered IP bi-weekly until euthanasia (criteria for euthanasia described above). As expected, lung metastases were observed in all mice. Despite the burden of disease in the lungs, TRC105 treatment resulted in a significant increase in survival (*p* = 0.0015, Figure 6B). Furthermore, no ovarian metastases were observed in TRC105-treated mice (0/8), while 78% of control mice (7/9) developed ovarian metastases. Similarly, while 78% (7/9) of control mice developed ascites, ascites were detected in only one mouse in the TRC105-treated group. Interestingly, this mouse had a tumor at the uterine horn without obvious ovarian metastasis. Significant reduction in liver metastasis was also observed in TRC105-treated mice: 37.5% (3/8) versus 100% (9/9) in controls (Figure 6C and Appendix A).

To study the effects of TRC105 on direct abdominal metastasis (to mimic ovarian cancer shedding), we injected OVCAR3 IP, followed by bi-weekly treatment with TRC105 at the same dose as above. As in the IP-injected mouse models using CD105-knockdown cells, TRC105-treated mice exhibited a significant survival advantage over controls (Figure 6D). In addition, TRC105-treated mice had a significant decrease in the number of abdominal tumor nodules (Figure 6E).

## 3. Discussion

We present here evidence indicating that CD105 plays an important role in mediating both metastasis to the ovary and abdominal spread. While CD105^(+)^ cells metastasize to the ovary, CD105-negative cells were unable to metastasize hematogenously to the ovary and, so, did not spread to the abdominal cavity. It is not that CD105^(−)^ cells were non-viable or could not tolerate being in the bloodstream, as they were able to lodge and grow in the lung.

The findings here are consistent with the literature. A recent publication by Zhang et al. identified a subset of paclitaxel-resistant CD105-expressing cancer stem-like cells that demonstrated increased metastatic capacity [44]. This study had a strength of linking CD105 with other CSC makers, while ours offers the strength of linking CD105 with potential early spread to the ovary. Further supporting a critical role for CD105 in ovarian metastasis, genetic knockdown of CD105 and pharmacologic targeting of CD105 with TRC105 both showed elimination of hematogenous ovarian metastasis. Interestingly, we did observe some hematogenous metastasis to the uterine horn in each scenario. Suggesting that CD105 plays a more general role in metastasis and not just spread to the ovary, pharmacologic inhibition with TRC105 also reduced the number of tumor implants with direct intraperitoneal injection of tumor cells. However, in this setting, TRC105 did not block spread to the ovary, suggesting CD105-mediated metastasis to the ovary is related to hematogenous spread.

Additional support for the role of CD105 in ovarian cancer metastasis comes from analysis of the ovarian cancer transcriptome; ovarian cancer cell lines and primary tumors that are highly enriched for CD105 demonstrate a high degree of enrichment for an overlapping set of signaling pathways related to cell adhesion and metastasis. Furthermore, knockdown of CD105 expression has been associated with a significant reduction in TGF-β receptors 1 and 2 and reduced phosphorylation of SMAD2/3. This was anticipated, given that CD105 is a TGF-β co-receptor, and is consistent with a large amount of literature linking the TGF-β superfamily to cancer cell epithelial mesenchymal transition and metastasis [35,45].

Our finding suggests that therapeutic targeting of CD105 could play an important role in preventing both hematogenous spread of ovarian cancer and implantation of direct peritoneal shedding. Anti-CD105 therapy and CD105 knockdown therapy were most efficacious in preventing hematogenous spread to the ovary. However, intraperitoneal therapeutic approaches also showed a reduction in the number of peritoneal nodules and improved overall survival in these mouse studies.

The ability to prevent metastasis could have an important therapeutic role in the prevention and treatment of established disease. Metastatic disease is a major clinical problem in ovarian cancer as most patients present with metastatic disease. However, once diagnosed, metastasis continues to be a major clinical problem of ovarian cancer patients; recurrent ovarian cancer continues to metastasize throughout the course of the disease, spreading throughout the peritoneal cavity, coating the bowels and mesentery, resulting in tumor ileus and/or bowel obstruction and significant morbidity and, ultimately, mortality [46,47]. The ability to prevent or delay metastasis could therefore improve survival and reduce patient symptoms.

Increasing evidence suggests that the majority of ovarian HGSCs arise in the fallopian tubes and then metastasize to the ovary [15,16]. Previous studies are consistent with those shown here and demonstrate that prevention of spread to the ovary significantly restricts abdominal metastasis [13,48,49]. Thus, antimetastatic agents could play a preventive role in patients at high risk of disease who are not candidates for prophylactic salpingectomy +/- oophorectomy.

This study also has implications for the cancer stem cell model of ovarian cancer. CD105^(+)^ cells were uniquely capable of metastasizing hematogenously to the ovary. However, CD105^(+)^-originated metastatic tumors did not reconstitute all the cell types found in the primary cell line, suggesting these cells are not multipotent stem-like cells, but rather progenitor cells capable of unlimited self-renewal. While these cells do express the CSC marker ALDH, they do not express many other reported CSC markers. In contrast, CD105-negative cells were unable to metastasize hematogenously but were able to lodge and grow in the lung. A population of CD105-positive cells was detected within the lung of mice injected with CD105-negative tumor cells. Thus, CD105^(−)^ cells could be a more stem-like population that differentiated to give rise to CD105^(+)^ cells. Alternatively, this could be due to FACS contamination of CD105^(−)^ cells, dedifferentiation, or trans-differentiation events. Additional studies will be necessary to evaluate the role of CD105 in an ovarian cancer differentiation hierarchy.

## 4. Materials and Methods 

### 4.1. Human Tumor Samples

All tumor samples were collected with IRB approval (PRO17090380, use of primary specimens from ovarian cancer patients for therapeutic development, approval date: 13 October 2017). Samples were taken from surgical blocks with clinically confirmed STIC lesions (p53 and Ki67 IHC). Samples included isolated STICs and samples with STIC and invasive tumors. Samples were obtained via the Magee-Womens Hospital and the Pitt Biospecimen Core (http://www.pittbiospecimencore.pitt.edu/) or the Penn OCRC Biotrust Collection (https://www.med.upenn.edu/OCRCBioTrust/).

### 4.2. Cell Culture and Antibodies

CAOV3, SKOV3, and HEY1 lines were purchased from ATCC. A2780 was provided by Dr. S. Murphy (Duke University), Kuramuchi was provided by Dr. Deborah Marsh (University of Sydney). OVCAR3 was proved by Dr. Kathleen Cho (University of Michigan). Ovarian cancer cell lines were cultured in Roswell Park Memorial Institute (RPMI)-1640 Media, supplemented with 10% fetal bovine serum (FBS), as previously described [13]. Anti-CD105 antibodies were purchased from Abcam (APC-conjugated, for FACS analysis) and eBioscience (SN6, for in vitro cell blocking). Smad2/3 and phospho-Smad2/3 antibodies were purchased from Cell Signaling Technology; GAPDH was purchased from Proteintech.

### 4.3. CD105 shRNA Knockdown

The pLKO CD105 shRNA clones (TRCN0000083138, TRCN0000083142) and scrambled control vectors (SHC002) were purchased from Sigma-Aldrich. Lentivirus with individual clones were produced in HEK 293T cells, using the psPAX2 and pMD2.G packaging systems. OVCAR3 cells and FACS-sorted CD105^(+)^ SKOV3 cells were transduced with lentivirus, then treated with Puromycin (1 μg/mL) to select the infected cells.

### 4.4. ECMatrix Cell Migration Assay

This assay was carried out per the manufacturer’s instructions (Millipore). To test the impact of CD105 knockdown (KD), control, or CD105 knockdown cells (1 × 10^5^) were seeded in the upper chamber in cell-free medium in triplicate. Anti-CD105 antibody (SN6, 4 µg/mL) or mouse IgG control was added to OVCAR3 in the upper chamber. FBS was added to the lower chamber, to a final concentration of 10%, as a chemoattractant. Cells that migrated out of the upper chamber or invaded the lower chamber after a 24 h incubation were lysed and subjected to fluorescent reading at 480–520 nm using a Gemini microplate reader (Molecular Devices).

### 4.5. CD105^(−/+)^ Cell Isolation

To ensure cellular purity while maintaining high levels of cellular viability, we used a two-step cell purification process. CD105^(−)^ and CD105^(+)^ SKOV3 cells were first separated using CD105 MicroBeads (MACS) per the manufacturer’s protocol. Each population was further stained with APC-conjugated CD105 antibody (Abcam), followed by FACS to purify CD105^(+)^ cells.

### 4.6. Cell Growth and Proliferation Assay

To compare cell proliferation rate, FACS-sorted CD105^(−)^ and CD105^(+)^ cells or cells with CD105 knockdown and corresponding scrambled controls were plated in 96-well plates (3000/well) in triplicate. At days 3, 6, and 9, the cells were incubated with the PrestoBlue reagent (Thermo Fisher) for 45 min, the fluorescence values were obtained using an Infinite 200 PRO microplate reader at Ex 560 nm and Em 590 nm.

### 4.7. Immunohistochemistry (IHC), Flow Cytometry, Immunoblotting, and qRT-PCR

Assays were carried out as previously described [12]. Paraffin sections of the fallopian tubes of 11 HGSC patients were used for IHC staining using anti-CD105 antibody (Sigma). Anti-p53 antibody staining (Proteintech) was used to identify STIC lesions, marked by strong nuclear staining. For fluorescence-activated cell sorting (FACS), APC-conjugated CD105 antibody (Abcam) was used. Antibodies used for immunoblotting were phosho-Smand2/3, total SMAD2/3 (Cell Signaling), and GAPDH (Proteintech). Primers for qRT-PCR were as follows: 

TGFBR1-RTF;ACGGCGTTACAGTGTTTCTG;

TGFBR1-RTR;GCACATACAAACGGCCTATCTC;

TGFBR2-RTF;GTAGCTCTGATGAGTGCAATGAC;

TGFBR2-RTR;CAGATATGGCAACTCCCAGTG;

TGFBR3-RTF;TGGGGTCTCCAGACTGTTTTT;

TGFBR3-RTR;CTGCTCCATACTCTTTTCGGG;

CD105-h-RTF;CACGCTCCCTCTGGCTGTTG;

CD105-h-RTR;CCCACAGGCTGAAGGTCACA;

HPRT-RTF;ATGCTGAGGATTTGGAAAGG;

HPRT-RTR;CAGAGGGCTACAATGTGATGG.

### 4.8. Mouse Model of Ovarian Cancer

All tumor studies were performed with the approval of either the University of Michigan (PRO00006094) or University of Pittsburgh IACUC (IS00011682). For each tumor cell type, mice were injected either subcutaneously (SQ) or via intravenous (IV) tail vein injection as indicated in Figure 1B. For CD105^(−/+)^-SKOV3 IV models, CD105^(−/+)^-sorted SKOV3 cells were used in tail vein injections (0.5–1 × 10^6^/mouse; *n* = 5 CD105^(−)^ and *n* = 8 CD105^(+)^). To study CD105 inhibition on ovarian metastasis, CD105 knockdown (ShCD105-1 or -2) or scrambled controls were used (1 × 10^6^/mouse; *n* = 5 each group). For HGSC line OVCAR3 IV models, OVCAR3 (0.5–1 × 10^6^/mouse; *n* = 7 each) were injected. For intraperitoneal (IP) tumor nodule growth, OVCAR3 shCD105 knockdown (shCD105-1 or -2) and scrambled controls (2 × 10^5^/mouse; *n* = 5 each) were injected into the abdomen as previously described [12].

### 4.9. Therapeutic Studies

OVCAR3 tumor cells were injected IV (0.5–1 × 10^6^/mouse; *n* = 9 for control and *n* = 8 for TRC105 treatment groups, respectively) or IP (1 × 10^5^/mouse; *n* = 10 each in control and TRC105 treatment groups) and treated with TRC105, 15 mg/kg IP, or PBS bi-weekly. For prevention of hematogenous spread in the IV tumor model, the first TRC105 treatment was administered 3 days prior to tumor cell injection. In the IP model, the first TRC105 treatment was administered at the time of tumor cell injection.

### 4.10. Pathway Enrichment Analysis of Ovarian Cancer Cell Lines and TCGA Ovarian Cancer Dataset

The ovarian cancer RNA-Seq dataset from TCGA was obtained from cBioPotal (OVCA-TCGA, http://www.cbioportal.org/). RNA-Seq data of ovarian cancer cell lines were obtained from the Cancer Cell Line Encyclopedia (OVCA-CCLE, https://portals.broadinstitute.org/ccle/). Quantile distribution (Q1, Q2, Q3 equals to 2.5, 5, 7.5, respectively) was obtained based on normalized CD105 expression (Z score). Genes expressed in CD105-high (Z > Q3) and CD105-low (Z < Q1) were compared in OVCA-CCLE. Gene expression fold changes between CD105-high and CD105-low were measured. Genes that had significant changes between these two groups (*p* < 0.05, average fold change > 1.5 or < 0.6) were used for pathway enrichment analysis using Crosstalker [50]. Gene co-expression was analyzed between CD105-high (Z > Q3) and the remining samples in OVCA-TCGA. Genes positively correlated (Pearson score *r* > 0.3, *p* < 0.05), or negatively correlated (*r* < -0.2, *p* < 0.05) with CD105-high were queried separated for pathway enrichment analysis using Crosstalker. 

### 4.11. Statistical Analysis

Student’s t-test was used for two-sample comparisons. Two-way ANOVA was used for group comparisons. The Gehan–Breslow–Wilcoxon test and Kaplan–Meier survival curve were used for survival analysis, *p* < 0.05 was considered statistically significant.

## 5. Conclusions

We found that CD105 expression correlates with a metastatic phenotype and is essential for ovarian cancer cell hematogenous metastasis to the ovary and subsequent intraperitoneal dissemination. Therapeutic targeting of CD105 both prevented hematogenous metastasis to the ovary and significantly reduced direct peritoneal implantation, thereby improving survival in murine models of ovarian cancer. This work supports the use of CD105-targeted agents as a new therapeutic approach in ovarian cancer.

## Figures and Tables

**Figure 1 cancers-11-01710-f001:**
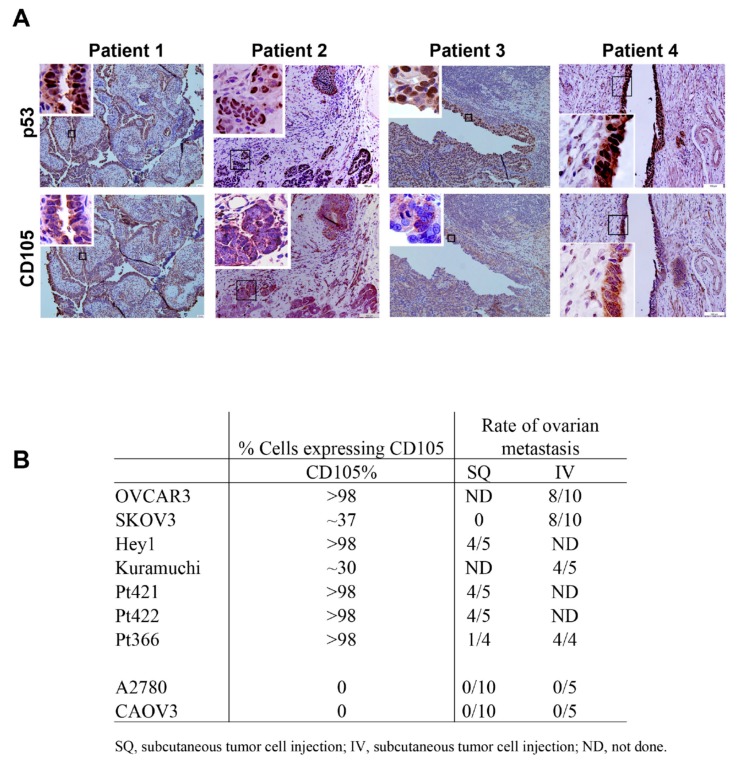
CD105 expression in fallopian tube serous tubal intraepithelial carcinoma (STIC) lesions and hematogenous ovarian metastasis of CD105-expressing cancer cells. (**A**) p53 IHC confirming strong nuclear p53 expression in STIC lesions, and CD105 IHC demonstrating CD105 expression in STIC lesions. Boxed areas indicate regions of high-power insets. (**B**) Summary of CD105 expression and rate of intravenous metastasis to the ovary of the indicated ovarian cancer cell lines and primary patient-derived cells.

**Figure 2 cancers-11-01710-f002:**
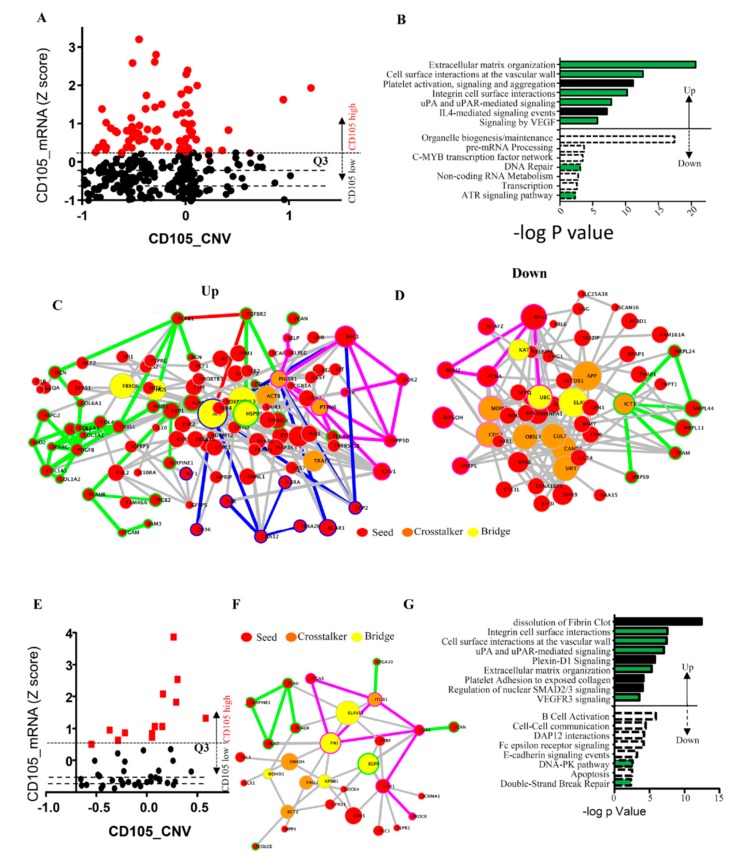
Functional protein interaction networks enriched in CD105-high ovarian cancer specimens and cell lines. (**A**) mRNA quantile distribution of CD105 expression in a 307-sample TCGA ovarian cancer RNA-Seq dataset (OVCA-TCGA, cBioPortal). Values at Q1–Q3 are indicated by dotted lines. CD105 high-expressing tumors are highlighted in red. (**B**) List of pathways enriched (Up) or reduced (Down) for CD105-high cells using Crosstalker network analysis (-log_10_
*p* value). (**C**–**D**) Protein network enriched (C) or reduced (D) in CD105-high ovarian cancer samples. (**E**) mRNA quantile distribution of CD105 expression in ovarian cancer cell lines from the CCLE RNA-Seq dataset (OVCA-CCLE). Values at Q1–Q3 are indicated by doted lines. CD105 high-expressing tumor cells are highlighted in red. (**F**) Protein network enriched in CD105-expressing ovarian tumor cell lines. (**G**) Significantly enriched or reduced pathway modules in the Crosstalker network analysis (-log_10_
*p* value). Green bars represent common pathways enriched/downregulated in both TCGA and CCLE datasets. Seeds (in Red) represent input gene list. Bridge (Yellow) and Crosstalker (Orange) represent non-seed genes that are significant in the particular enriched network.

**Figure 3 cancers-11-01710-f003:**
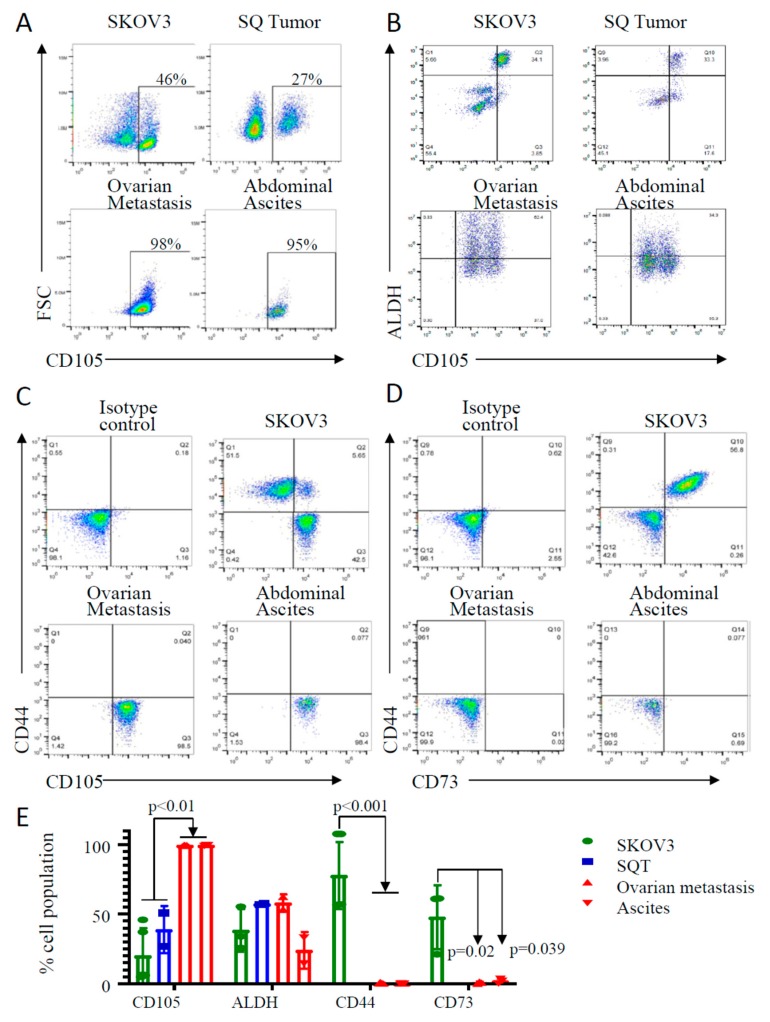
Cell surface marker and cancer stem cell marker expression in ovarian and abdominal metastases. (**A**–**D**) Flow cytometric analysis of bulk SKOV3 cells in culture, tumor cells derived from subcutaneously injected SKOV3 (SQ) and ovarian metastasis and abdominal metastasis from IV-injected SKOV3 for the indicated cell markers. Results are representative analyses from two to three independent experiments. (**E**) Summary of CD105 and other stem cell markers expression in SQ tumor, ovarian metastasis, abdominal metastasis, and bulk SKOV3 ovarian cancer cell line as described in (**A**–**D**).

**Figure 4 cancers-11-01710-f004:**
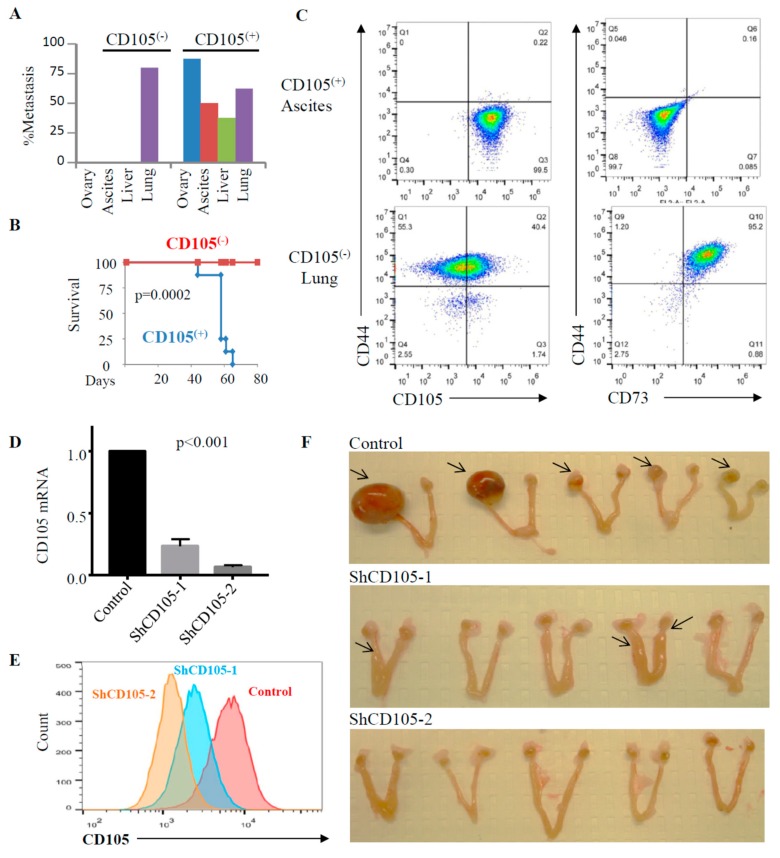
CD105^(+)^ cells, but not CD105^(−)^ cells, metastasized to the ovary hematogenously, which can be prevented by CD105 knockdown in SKOV3 cells. (**A**) Percent of animals developing metastases at the indicated sites from intravenously injected CD105^(+)^ and CD105^(−)^ cells. Metastasis was recorded in the indicated tissues at time of euthanasia. (**B**) Kaplan–Meier survival curves of mice injected by IV with CD105^(−)^ or CD105^(+)^ SKOV3 tumor cells (*n* = 5/group for CD105^(−)^, *n* = 8/group for CD105^(+)^). (**C**) Flow cytometric analysis of expression of the indicated cell markers of CD105^(+)^ cell-derived metastatic ascites and CD015^(−)^ cell-derived lung tumors. (**D**,**E**) CD105 knockdown eliminates hematogenous ovarian metastasis of SKOV3 cells. (**D**) qRT-PCR evaluation of CD105 mRNA expression in scrambled shRNA control and CD105 shRNA knockdown cells in FACS-sorted CD105^(+)^ SKOV3 cells. The level of CD105 in control cells is defined as 1 after normalization with HPRT. The values are the means ± SD from three independent experiments (*p* < 0.001). (**E**) Flow cytometric analysis of CD105 protein expression in scrambled shRNA control, shCD105-1, and shCD105-2 in FACS-sorted CD105^(+)^ SKOV3 cells. (**F**) Images of ovaries and uterine horn metastasis in the indicated control and CD105-knockdown cell lines (*n* = 5 each). Mice were sacrificed 4 weeks after cell injection.

**Figure 5 cancers-11-01710-f005:**
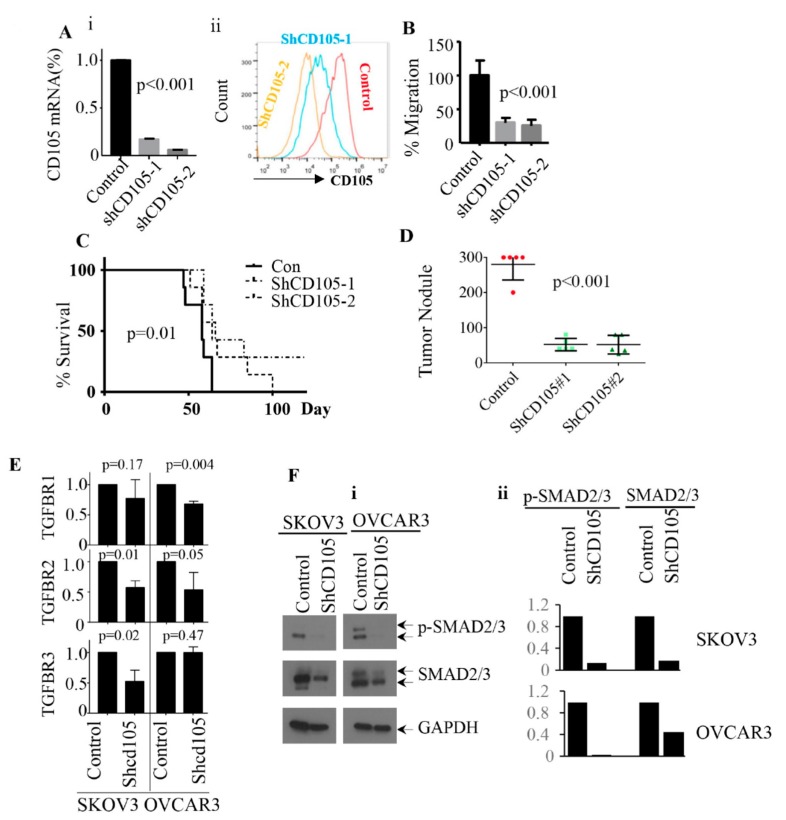
CD105 knockdown reduced ovarian metastasis and improved survival in mouse models of high-grade serous ovarian cancers (HGSC) ovarian cancer cells. (**A**) i-ii qRT-PCR and flow cytometry analysis of CD105 mRNA and protein expression in control, shCD105-1, and shCD105-2 OVCAR3 cells. CD105 mRNA in control cells was set as 1 after normalization with HPRT. The values are the means ± SD from three independent experiments (*p* < 0.001). (**B**) Summary of average cell migration potential, from three independent experiments, of CD105-knockdown cells relative to control cells (control cell migration defined as 100%). (**C**) Kaplan–Meier survival curves of mice injected with control, shCD105-1, or shCD105-2 (*n* = 7 each) bearing IV-injected OVCAR3 ovarian tumor cells. (**D**) Summary of intra-abdominal tumor nodules in mice intraperitoneally injected with control and CD105-knockdown cells (*n* = 5 per group, *p* < 0.001). For quantification, tumor nodule numbers over 300 were estimated as 300. (**E**) mRNA expression of the indicated TGF-β receptors in control and CD105-knockdown SKOV3 and OVCAR3 cells; expression was defined as 1 after normalization with HPRT. The values are the means ± SD from three independent experiments (*p* values are as indicated) (**F**) (i) Western blot analysis of phosphorylated SMAD2/3 (p-SMAD2/3) and total SMAD2/3 in control and CD105 knockdown SKOV3 and OVCAR3 cells and (ii) densitometry analysis of Western blot using Image J. Expression value of SMAD2/3 and (p-SMAD2/3) in control cells were set as 1 after normalization with the corresponding GAPDH level.

**Figure 6 cancers-11-01710-f006:**
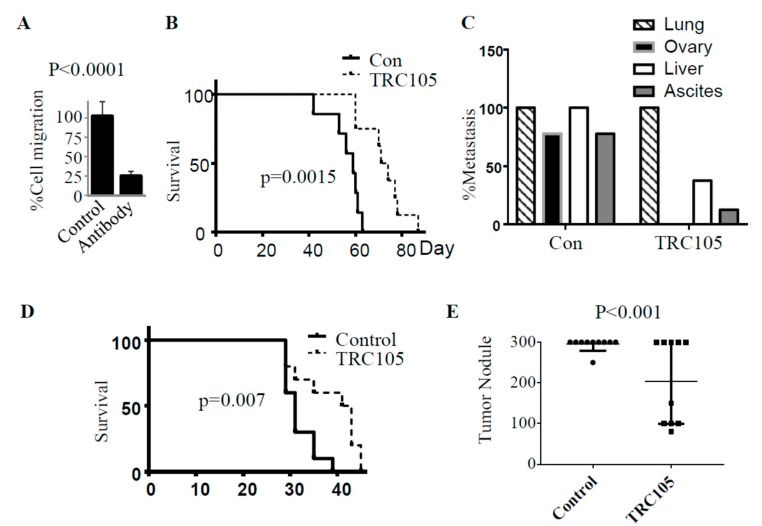
Impact of CD105 neutralizing antibody on ovarian metastasis and survival. (**A**) Summary of OVCAR3 cell migration in the presence or absence of CD105-neutralizing antibody. The experiment was repeated three times. Migration was normalized to that of control cells, defined as 100%. (**B**) Kaplan–Meier survival curve of mice injected intravenously with OVCAR3 cells and treated with control vehicle (*n* = 9) or TRC105 CD105-neutralizing antibody (*n* = 8). (**C**) Metastasis incidence in the indicated organs in control and TRC105-treated mice with IV-injected OVCAR3. (**D**) Kaplan–Meier survival curve of mice injected intraperitoneally (IP) with OVCAR3 cells and treated with control vehicle (*n* = 10) or TRC105 CD105-neutralizing antibody (*n* = 10). (**E**) Average number of abdominal tumor implants in control and TRC105-treated mice with IP tumor injection (*N* = 10 per group, *p* < 0.001). For quantification, tumor nodule numbers over 300 were estimated as 300.

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
