# Peer review of "CD105 Is Expressed in Ovarian Cancer Precursor Lesions and Is Required for Metastasis to the Ovary"

_cancers, 2019, doi:10.3390/cancers11111710_

Round 1
Reviewer 1 Report
This study describes that CD105 expression correlates with a metastatic phenotype and is essential for ovarian cancer cell hematogenous metastasis and is interesting. It will be acceptable for the publication of "Cancers" if several points are revised as below.
It will be better to describe an official name, endoglin (ENG) rather than general name, CD105 in this paper (Abstract, keywords, etc.) Fig2, in line 123, please correct CD105-Highigh-expressing to CD105-high-expressing.Fig3.. Please add statistics & A, B, C, D legends. Fig4D, Please add x-xis and y-xis. In line 366, please add IRB number. In line 380, please check the concentration of puromycin. It will be better to add a recent paper (Human epithelial ovarian cancer cells expressing CD105, CD44 and CD106 surface markers exhibit increased invasive capacity and drug resistance. Oncol Lett. 2019 Jun;17(6):5351-5360. doi: 10.3892/ol.2019.10221) and compare your findings (strong points and difference) with it in discussion section.
Author Response
Reviewer#1,
It will be better to describe an official name, endoglin (ENG) rather than general name, CD105 in this paper (Abstract, keywords, etc.)
We have added official name “endoglin” and “ENG” in abstract, introduction and keyword sections as requested
Fig2, in line 123, please correct CD105-Highigh-expressing to CD105-high-expressing.
We apologize for this typo. This has been corrected.
. Please add statistics & A, B, C, D legends and x-xis and y-xis were in Fig4D.
We apologize the legend was somehow cut-off with the figure submission. The figure has now been cropped to ensure they are visualized. Statistics are difficult to show on a single FACS plot. We have therefore added and presented in a new bar graph in Fig. 3E which summarizes results across multiple samples.
Please add IRB number, name and approval dates, check puromycin levels
These have been added, and the puromycin level corrected.
It will be better to add a recent paper (Human epithelial ovarian cancer cells expressing CD105, CD44 and CD106 surface markers exhibit increased invasive capacity and drug resistance. Oncol Lett. 2019 Jun;17(6):5351-5360. doi: 10.3892/ol.2019.10221) and compare your findings (strong points and difference) with it in discussion section.
We appreciate the reviewer’s excellent observation and have added this reference as well as addressed this in the discussion as suggested.
Reviewer 2 Report
Here author studied the role of CD105 as a critical mediator of ovarian metastasis.
I have some comments:
Author fails to explain why CD105+ cancer cells metastasized to lung. Even antibody treatment failed to stop metastasis to lung?
Any other mouse model such as syngenic spontaneous mouse model or PDX model would have created more impact than experimental metastasis model
Author Response
Response to Reviewer 2:
Author fails to explain why CD105+ cancer cells metastasized to lung. Even antibody treatment failed to stop metastasis to lung?We used tail vein IV-injection of ovarian cancer cells as model to study ovarian homing and metastasis. As the lung acts as a filter organ in this model, tumor cells that are trapped in the lung microvasculature can grow into tumors without requiring intravasation as in other tumors as such this many not represent true metastasis. We discussed this extensively in our prior publication of the metastasis models (Coffman et. al. Translational Research).
Any other mouse model such as syngenic spontaneous mouse model or PDX model would have created more impact than experimental metastasis model.
We agree that other mouse model such as syngeneic spontaneous mouse model such as parabiosis or PDX model would have created more impact than the IV experimental metastasis model. However these models are much more time consuming and would have required a cost-prohibitive amount of anti-CD105 antibody and thus were not feasible. Similarly given the tight timeline for resubmission, they are not possible with resubmission.